# Bag-1 Protects Nucleus Pulposus Cells from Oxidative Stress by Interacting with HSP70

**DOI:** 10.3390/biomedicines11030863

**Published:** 2023-03-12

**Authors:** Kaori Suyama, Daisuke Sakai, Shogo Hayashi, Ning Qu, Hayato Terayama, Daisuke Kiyoshima, Kenta Nagahori, Masahiko Watanabe

**Affiliations:** 1Department of Anatomy and Cellular Biology, Basic Medical Science, Tokai University School of Medicine, 143 Shimokasuya, Isehara 259-1193, Kanagawa, Japan; sho5-884@umin.ac.jp (S.H.); quning@tokai.ac.jp (N.Q.); terahaya@tokai-u.jp (H.T.); kiyoshima@tokai.ac.jp (D.K.); nk761111@tsc.u-tokai.ac.jp (K.N.); 2Department of Orthopaedic Surgery, Surgical Science, Tokai University School of Medicine, 143 Shimokasuya, Isehara 259-1193, Kanagawa, Japan; daisakai@is.icc.u-tokai.ac.jp (D.S.); masahiko@is.icc.u-tokai.ac.jp (M.W.)

**Keywords:** Bag-1, nucleus pulposus, oxidative stress, HSP70

## Abstract

Bcl-2-associated athanogene 1 (Bag-1) is a multifunctional prosurvival protein that binds to several intracellular targets and promotes cell survival. HSP70 and Raf-1 are important targets of Bag-1; however, the protective function of Bag-1 in nucleus pulposus (NP) cells remains unclear. In this study, we determined the effects of Bag-1 on NP cells under oxidative stress induced by treatment with hydrogen peroxide (H_2_O_2_). We found that Bag-1 was bound to HSP70, but Bag-1–Raf1 binding did not occur in NP cells. Bag-1 overexpression in NP cells enhanced cell viability and mitochondrial function and significantly suppressed p38/MAPKs phosphorylation during oxidative stress, although NP cells treated with a Bag-1 C-terminal inhibitor, which is the binding site of HSP70 and Raf-1, decreased cell viability and mitochondrial function during oxidative stress. Furthermore, the phosphorylation of the ERK/MAPKs was significantly increased in Bag-1 C-terminal inhibitor-treated NP cells without H_2_O_2_ treatment but did not change with H_2_O_2_ exposure. The phosphorylation of Raf-1 was not influenced by Bag-1 overexpression or Bag-1 C-terminal binding site inhibition. Overall, the results suggest that Bag-1 preferentially interacts with HSP70, rather than Raf-1, to protect NP cells against oxidative stress.

## 1. Introduction

The intervertebral disc (IVD) is an organ that exists between the vertebrae. It is composed of an outer fibrocartilaginous annulus fibrosus (AF) that surrounds a gel-like nucleus pulposus (NP). The IVD is an avascular tissue that exchanges nutrients and metabolites primarily by diffusion to and from microvessels in the superior and inferior cartilaginous end plate of the vertebrae and outer AF [1,2,3]. The NP consists of a small number of NP cells scattered in the extracellular matrix, which absorb shock and maintain spinal mobility. NP cells have adapted to survive in this unique, hypoxic environment [4]. As a result of aging, mechanical stress, genetics, and many other factors, the number of functional NP cells decreases through apoptosis and cellular senescence. This is accompanied by a phenotypic shift toward catabolism and results in gradual IVD degeneration. IVD degeneration is clinically related to disc herniation, spinal canal stenosis, spinal deformities, and chronic low back pain, which has a profound effect on patient quality of life and causes a considerable socioeconomic burden [1,2]. Recent studies have shown that oxidative stress resulting from excessive reactive oxygen species (ROS) promotes apoptosis, necrosis, and senescence of NP cells in the disc microenvironment [5,6,7,8,9]. Excessive ROS can induce mitochondrial dysfunction, including mitochondrial membrane potential collapse, ultrastructure disintegration, and ATP depletion. This results in additional ROS accumulation as well as DNA damage, cell senescence, and disruption of NP homeostasis [10,11,12].

Bcl-2-associated anthanogene 1 (Bag-1) is a multifunctional, prosurvival protein that binds to Bcl-2 and promotes cell survival [13]. However, the predominant function of Bag-1 was reported to be a co-chaperone for the 70 kDa heat shock protein (HSP70) to activate and regulate its protective function [14,15,16]. The Bag-1 gene encodes three major protein isoforms in human cells: Bag-1L (52 kDa), Bag-1M (46–48 kDa) and Bag-1S (36 kDa) in human cells, and 32 kDa (Bag-1S) and 50 kDa (Bag-1L) isoforms in mouse cells [17,18]. All Bag-1 isoforms share a highly conserved protein interaction region near the C-terminus, which binds to the HSP70/heat shock cognate 70 (HSC70) molecular chaperone to regulate its ATPase activity, which is closely related to the anti-apoptotic function of HSP70 [14,15,19,20]. Several studies indicate that the function of HSP70 is to protect cells from oxidative stress-related damage, not just heat shock, and to regulate mitochondrial function [6,10,11,21,22,23]. Moreover, Bag-1 interacts with the catalytic domain of the protein kinase Raf-1(c-Raf) to induce its activation independently from the influence of Ras [24,25,26]. HSP70 and Raf-1 compete for binding to the C-terminus of Bag-1 [15,27]. The effects of Bag-1 are also mediated by the activation of the Raf-1-dependent mitogen-activated protein kinase (MAPK) pathway [10,25,27], which contributes to the cellular response to oxidative stress, such as ROS-triggered apoptosis [21]. Additionally, HSP70 is one of the factors that influences MAPKs from various directions or phases [11,27,28,29]. Thus, previous studies regarding the relationship between Bag-1, HSP70, and Raf-1 suggest the possibility that crosstalk plays an important role in IVD degeneration. Although the effects of Bag-1, Bag-1-HSP70, and Bag-1-Raf-1 in various cells and tissues have been widely reported, the effects of Bag-1 on IVD cells, including NP and AF cells, and whether a relationship exists between Bag-1, HSP70, and Raf-1 in IVD cells is unclear. Therefore, we determined the effects of Bag-1 in NP cells under oxidative stress and the association of Bag-1 with HSP70 and Raf-1, for the first time.

## 2. Materials and Methods

### 2.1. Isolation of NP Cells, Hypoxic Culture Conditions, and Cell Treatments

Rat NP cells were isolated from male Sprague-Dawley rats (11 weeks old) using a modified method described by Risbud et al. [30]. Briefly, IVDs from the lumbar and coccygeal regions were dissected from rats under deep aesthesia using aseptic conditions. The gel-like NP tissue was separated from the AF and the NP tissue was minced by pipetting. The isolated cells were maintained in Dulbecco’s modified Eagle’s medium (DMEM) (Nakarai Tesque, Kyoto, Japan) supplemented with 10% fetal bovine serum (Gibco, Grand Island, NY, USA) antibiotics and cultured in a Hypoxia Chamber (MIC-101; Billups Rothenberg Inc., Del Mar, CA, USA) containing 5% CO_2_, 94% N_2_, and 1% O_2_, which reflects the in vivo environment. The NP cells were seeded into dishes and incubated in DMEM at 37 °C until approximately 90% confluence was reached on the following day. The medium was replaced and was supplemented with or without hydrogen peroxide (H_2_O_2_). Thioflavin-S (NSC71948; Sigma-Aldrich, St. Louis, MO, USA) is a small molecule, chemical inhibitor of the interaction between Bag-1 and heat shock proteins, including HSC70, HSP70, and Raf-1 kinase [31,32]. NP cells were treated with 100 μM thioflavin-S at approximately 80% confluent for 16 h and the medium was replaced and supplemented with or without H_2_O_2_.

### 2.2. Bag-1 Overexpression

The pIRES2-AcGFP1 plasmid (Clontech Laboratories, Mountain View, CA, USA), contains the internal ribosome entry site of the encephalomyocarditis virus between the multiple cloning site, the kanamycin/neomycin resistance gene, and the Aequorea coerulescens green fluorescent protein (AcGFP) coding region. The cDNA of rat Bag-1 (GenBank NM_001106647.3) covering the entire open reading frame was cloned into the pIRES2-AcGFP1 plasmid. NP cells were seeded into 6-well plates at approximately 80% confluence 24 h before transfection. NP cells were transfected with the Bag-1 expression plasmid using Lipofectamine 3000 (Invitrogen, Carlsbad, CA, USA) reagent. Data were collected by BD LSR Fortessa (Beckton Dickinson, San Jose, CA, USA) and analyzed with FlowJo version 10.8.1 software (Beckton Dickinson). From after 24–48 h of transfection, the cells were incubated with 800 μg/mL of G418 (Nacalai Tesque)-containing medium for selection from over 7–10 days. The transfected cells were seeded onto new plates and used for subsequent experiments.

### 2.3. Immunohistological Studies

To gain insight into the expression of Bag-1 in the IVD, freshly isolated spines from 11-week-old rats were fixed in 4% paraformaldehyde in phosphate-buffered saline (PBS), decalcified and embedded in paraffin wax. Sagittal sections were deparaffinized in xylene, rehydrated through a graded ethanol series and incubated with antibodies against Bag-1 (#ab32109; Abcam plc, Cambridge, UK) in 2% normal goat serum in tris buffered saline with Tween 20 (TBS-T) at a 1:50 dilution overnight at 4 °C. The sections were washed thoroughly and then stained with N-Histofine^®^ Simple Stain™ Rat MAX PO (NICHIREI BIOSCIENCES INC. Tokyo, Japan). Nuclei were stained with haematoxylin. The color was developed using 3, 30-diaminobenzidine (Vectastain Universal Quick Kit; Vector Laboratories Inc., Newark, CA, USA) and the slides were examined under a microscope (BX63; Olympus, Tokyo, Japan).

### 2.4. Cell Viability Assay

The NP cells were seeded into 96-well plates and incubated in DMEM at 37 °C at approximately 90% confluence, treated with 200 and 400 μM of H_2_O_2_ in DMEM medium, and incubated 37 °C for 24 h. Cell viability was evaluated using the Cell Counting Kit-8 assay (CCK-8; Dojindo, Kumamoto, Japan) following the manufacturer’s protocol. Measurements at 450/650 nm were made 2 h after the addition of CCK-8 reagent using a SpectraMax i3 Multi-Mode Microplate Reader (Molecular Devices, San Jose, CA, USA).

### 2.5. Measurement of Intracellular ROS

Intracellular ROS levels were assessed using fluorogenic probes with the ROS Assay Kit—Photo-oxidation Resistant DCFH-DA—(R253; Dojindo) following the manufacturer’s protocol. The cells were incubated for 30 min at 37 °C with the ROS Assay working solution and then treated with H_2_O_2_. After washing twice with DMEM, the cells were fixed with 4% paraformaldehyde and Hoechst 33342 (Dojindo) was added to detect the nuclei. The stained cells were observed by fluorescence microscopy (Axio Imager M2; Carl ZEISS AG, Oberkochen, Germany) at 550/605nm (excitation/emission). The fluorescence intensity derived from DCFH-DA was calculated as the average intensity value of the stained cells by ZEISS ZEN 3.2 (blue edition) software (Carl ZEISS AG).

### 2.6. Detection of Mitochondrial Membrane Potential

To assess mitochondrial function, the mitochondrial membrane potential was measured using the MT-1 MitoMP Detection Kit (MT-1; Dojido) following the manufacturer’s protocol. Cells were incubated with MT-1 working solution for 30 min at 37 °C, followed by treatment with H_2_O_2_. After two washes with DMEM, the cells were fixed with 4% paraformaldehyde and Hoechst 33342 was added to detect the nuclei. The stained cells were observed by fluorescence microscopy (Axio Imager M2; Carl ZEISS AG) at 470/525nm (excitation/emission). The fluorescence intensity derived from MT-1 was calculated as the average intensity value of the stained cells by ZEISS ZEN 3.2 (blue edition) software (Carl ZEISS AG).

### 2.7. Real-Time RT-PCR Analysis

Total RNA was extracted from NP cells using RNAeasy mini columns (Qiagen, Hilden, Germany). Before elution, the RNA was treated with RNase-free DNase I (Qiagen). The purified, DNA-free RNA was converted to cDNA using the High-Capacity cDNA Reverse Transcription Kits (Applied Biosystems, Foster City, CA, USA). Template cDNA and gene-specific primers were added to Power SYBR Green Master Mix (Applied Biosystems) and mRNA expression was quantified using the Step One Plus Real-time PCR System (Applied Biosystems). β-actin was used to normalize gene expression. Melting curves were analyzed to verify the specificity of the PCR product and the absence of primer-dimers.

### 2.8. Protein Extraction, Western Blot Analysis, and Immunoprecipitation

At the indicated time points following treatment, the cells were placed on ice and washed with ice-cold PBS. The cells were lysed with RIPA Buffer (#9806; Cell Signaling Technology, Danvers, MA, USA) containing cOmplete^TM^ Protease Inhibitor Cocktail (Roche Diagnostics, Mannheim, Germany) and PhosSTOP^TM^ (Roche). Immunoprecipitation was performed using Protein A Agarose Beads (#9863; Cell Signaling Technology) following a standard protocol. Proteins were separated by sodium dodecyl sulfate-polyacrylamide gel electrophoresis and transferred to Immobilon-P polyvinylidene difluoride membranes (Millipore, Billerica, MA, USA). The membranes were blocked with blocking buffer (5% bovine serum albumin, 0.1% NaN_3_ in PBS) and incubated overnight at 4 °C with antibodies against Bag-1 (#ab32109; Abcam plc), p38 (#8690; Cell Signaling Technology), phosphorylated p38 (#4511; Cell Signaling Technology), ERK (#4695; Cell Signaling Technology), phosphorylated ERK (#4370; Cell Signaling Technology), JNK (#9252; Cell Signaling Technology), phosphorylated JNK (#AF1205; R&D Systems, Minneapolis, MN, USA), β-actin (#A2228; Sigma-Aldrich), or GAPDH (#G9545; Sigma-Aldrich). All antibodies were diluted in Can Get Signal Immunoreaction Enhancer Solution (Toyobo, Tokyo, Japan). Chemiluminescence signals were visualized with Immobilon Western Chemiluminescent HRP Substrate (Millipore) or VeriBlot for IP Detection Reagent (HRP) (#ab131366; Abcam) and scanned using a Vilber Bio Imaging FUSION (M&S Instruments Inc., Osaka, Japan). Western blot data were quantified by densitometric scans of the films using the computer software Evolution-Capt Edge (M&S Instruments Inc.) and presented as band intensities normalized to that of the loading control (β-actin or GAPDH).

### 2.9. Statistical Analysis

All measurements were performed at least three times and the data are presented as the mean ± standard deviations (SD). Differences between groups were analyzed using the Student’s *t*-test or Tukey–Kramer test. Statistical significance was set at *p* < 0.05.

## 3. Results

### 3.1. Evaluation of Bag-1 Expression in NP Cells

To confirm the expression of Bag-1 in NP tissue, we stained sections of rat discs with antibodies against Bag-1 (Figure 1A,B). The results showed prominent expression of Bag-1 in NP tissue with many cells.

The average transfection efficiency of Bag-1 plasmid into NP cells was 25.1% at 48 h after transfection (Figure 1C). Next, to measure the expression of Bag-1 in cells after selection, real-time PCR was used to quantitate Bag-1 mRNA in NP cells and in NP cells transfected with the Bag-1 gene. Bag-1 mRNA was significantly increased in Bag-1-transfected NP cells compared with untreated NP cells (Figure 1D). Similarly, quantitation of immunoblots prepared with Bag-1-transfected cell extracts revealed a significant increase in Bag-1 protein compared with untreated NP cells (Figure 1E). These findings indicate that Bag-1-transfected and antibiotic-selected NP cells stably express Bag-1 mRNA and these NP cells may be considered Bag-1-overexpressing cells compared with normal NP cells.

### 3.2. H_2_O_2_ Impaired NP Cell Viability and Mitochondrial Function and Increased Intracellular ROS

Cell viability was measured to determine the effects of H_2_O_2_ on NP cells from 0–400 μM. CCK-8 assays revealed that the groups treated with between 200 and 400 μM H_2_O_2_ for 24 h exhibited significantly decreased cell viability compared with the control (Figure 2A). The production of intracellular reactive oxygen spices (ROS) increased significantly with H_2_O_2_ treatment in a dose-dependent manner (Figure 2B,C). Conversely, the analysis of mitochondrial membrane potential using MT-1 dye revealed red fluorescence without H_2_O_2_ treatment, whereas H_2_O_2_ treatment resulted in a significant decrease in the intensity of MT-1 derived fluorescence (Figure 2B,D).

### 3.3. Bag-1-Overexpressing NP Cells Attenuate the Effect of H_2_O_2_ on Cell Viability, Mitochondrial Function, and Increased Intracellular ROS Levels

The viability of Bag-1-overexpressing NP cells by CCK-8 analysis did not show a significant change under 200 μM H_2_O_2_ treatment for 24 h compared with the untreated group, whereas a significant decrease was observed under 400 μM H_2_O_2_ treatment compared with the untreated group and 200 μM H_2_O_2_ treatment for 24 h (Figure 3A). The viability of Bag-1-overexpressing NP cells was significantly higher compared with that of NP cells treated with similar H_2_O_2_ concentrations (Figure 3B). The level of intracellular ROS production increased significantly with H_2_O_2_ treatment at 200 and 400 μM for 24 h compared with the untreated group (Figure 3C,D). Moreover, the mitochondrial membrane potential was not significantly altered in cells treated with H_2_O_2_ (Figure 3C,E).

### 3.4. Bag-1 Binds to HSP70 in NP Cells, but Does Exhibit Obvious Raf-1 Binding

Previous studies have demonstrated an interaction between Bag-1, HSP70, and Raf-1 in various cells [13,15,19,20,25,26,27,33]. Therefore, we evaluated this putative protein–protein binding in NP cells. NP cell extracts were immunoprecipitated with anti-Bag-1 antibody and the direct interaction partners were detected by immunoblotting using specific antibodies. As expected, Bag-1 was bound to HSP70 and these complexes were evident following treatment with between 200 and 400 μM H_2_O_2_ for 24 h (Figure 4A). Similarly, immunoprecipitation of HSP70 resulted in co-precipitation of Bag-1 under all conditions (Figure 4A). In contrast, immunoprecipitation of Bag-1 in NP cells did not result in co-precipitation of Raf-1 under all conditions (Figure 4B). Additionally, we only detected an exceedingly faint expression of Bag-1 by immunoprecipitation of Raf-1 following treatment with between 200 and 400 μM H_2_O_2_ (Figure 4B).

### 3.5. Treatment of Bag-1 with an Inhibitor of the Binding Site for HSP70 and Raf-1 Attenuates NP Cell Viability, Mitochondrial Function, and Increased Intracellular ROS Levels

Thioflavin-S (NSC71948) is an inhibitor of the interaction between Bag-1 and HSP70 or Raf-1 [31,32]. This compound selectively inhibits Bag-1 binding through the C-terminal binding site of HSP70 or Raf-1 [32]. To determine whether Bag-1-HSP70 binding is involved in protecting NP cells from oxidative stress, NP cells were incubated with thioflavin-S and cell viability, ROS accumulation, and mitochondrial activity were measured following H_2_O_2_ treatment. The cell viability of NP cells decreased markedly following treatment with between 200 and 400 μM H_2_O_2_ for 24 h (Figure 5A). The viability of NP cells was significantly higher compared with that of thioflavin-S-treated NP cells at similar H_2_O_2_ concentrations (Figure 5B). ROS accumulation in thioflavin-S-treated NP cells was increased significantly following between 200 and 400 μM H_2_O_2_ treatment for 24 h in a dose-dependent manner (Figure 5C,D). The mitochondrial membrane potential was significantly decreased based on the intensity of MT-1-derived fluorescence following H_2_O_2_ treatment (Figure 5C,E).

### 3.6. The Effects of Bag-1 on MAPKs and Raf-1 Activation

To further determine the effects of the putative Bag-1, HSP70, and Raf-1 interactions on MAPK kinase activity (p38, ERK1/2 and JNK), we measured the phosphorylation status of p38, ERK1/2, JNK, and Raf-1 following H_2_O_2_ treatment by Western blot analysis. NP cells were treated with between 200 and 400 μM H_2_O_2_ under hypoxic conditions for 30 min. The phosphorylation of Raf-1 did not increase significantly with 200 μM H_2_O_2_ treatment but increased significantly with 400 μM H_2_O_2_ treatment compared with 0 μM H_2_O_2_ (Figure 6A–C). The phosphorylation of p38 and ERK1/2 in NP cells increased significantly with between 200 and 400 μM H_2_O_2_ treatment compared with untreated cells (Figure 6A–C).

Similarly, in Bag-1-overexpressing cells, the phosphorylation of Raf-1 did not significantly increase at 200 μM H_2_O_2_ but increased significantly at 400 μM H_2_O_2_ compared with the untreated cells (Figure 7A–C). Unlike NP cells, the phosphorylation of p38 in Bag-1-overexpressing cells did not significantly change following treatment with between 200 and 400 μM H_2_O_2_ (Figure 7B). In Bag-1 overexpressing NP cells, the phosphorylation of ERK1/2 in-creased significantly following treatment with between 200 and 400 μM H_2_O_2_ compared with the untreated cells (Figure 7B,C). In untreated Bag-1-overexpressing NP cells, the phosphorylation of ERK1/2 did not significantly change compared with NP cells (Figure 7D).

In thioflavin-S-treated NP cells, the phosphorylation of p38 increased significantly following treatment with between 200 and 400 μM H_2_O_2_ treatment compared with untreated cells and the phosphorylation of Raf-1 increased significantly at 400 μM H_2_O_2_, although the phosphorylation of ERK1/2 did not change significantly under all conditions (Figure 8A–C). Without H_2_O_2_, the phosphorylation of ERK1/2 in thioflavin-S treated NP cells was significantly increased compared with NP cells (Figure 8D).

In addition, the phosphorylation of JNK in NP cells, Bag-1 overexpressing NP cells, and thioflavin-S-treated NP cells showed no significant changes under all conditions (Figure 6, Figure 7 and Figure 8B,C).

## 4. Discussion

Previous studies on the aging of vertebrate discs reported that H_2_O_2_ induces excessive ROS production, which leads to oxidative stress and damaged cells. Mitochondria also suffer damage and dysfunction as evidenced by the loss of mitochondrial membrane potential [5,8]. Mitochondrial dysfunction is associated with aging and degenerative disease progression, which results in cell damage as a secondary consequence of oxidative stress [34,35]. Several recent studies have reported that interventions targeting oxidative stress protect NP cells from damage and cell death, thus, preventing IVD degeneration [5,6,7,8,9].

The MAPK signaling pathway is an important pathway involved in ROS-triggered cell damage in NP cells [5,7,8,12,34,36]. In the present study, H_2_O_2_ treatment increased NP cell death and upregulated the phosphorylated p38 and ERK levels. It was reported that p38 activation is a marker of senescence and induces the senescence-associated secretory phenotype [2,37]. Other studies on NP cells have demonstrated H_2_O_2_-induced oxidative stress through MAPK signaling pathways, particularly p38 [5,12,38,39,40]. Thus, MAPK activation effectively suppresses the accumulation of ROS and induces cell senescence.

Bag-1 protects cells and tissues against various stresses, such as oxidative stress, by interacting with HSP70 in response to stressors [14,18,25,33]. HSP70 contains a nucleotide-binding domain (NBD) where ATP is bound and hydrolyzed [15]. The C-terminus of Bag-1 binds to the NBD of HSP70 and acts as a nucleotide-exchange factor; this interaction assists HSP70 function as a chaperone [15,33,41]. Bag-1 overexpression increases cell viability and protects cells by upregulating HSP70 activity [14,17,25], which suggests that Bag-1 stabilizes HSP70 protein in cells [14]. HSP70 regulates the quality of intracellular protein folding and contributes to cell survival under various conditions [41]. Furthermore, overexpression of HSP70 suppresses ROS production from the mitochondria [22] and regulates mitochondrial function to adapt NP cells to various stresses [10].

In the present study, H_2_O_2_ induced intracellular ROS levels in Bag-1-overexpressing NP cells, but mitochondrial function did not decrease, whereas NP cells treated with thioflavin-S exhibited increased ROS accumulation and decreased mitochondrial function and cell viability. In addition, our findings indicated that Bag-1 binds to HSP70, whereas we did not clearly observe binding between Bag-1 and Raf-1 under normal conditions or after H_2_O_2_ treatment. Raf-1 immunoprecipitation showed only a slight co-precipitation band under H_2_O_2_ treatment; however, it is possible that Raf-1–Bag-1 binding may gradually increase during stress intensity. Although it is difficult to precisely quantitate the amount of Bag-1–HSP70 and Bag-1–Raf-1 binding, our results indicate that Bag-1 preferentially binds to HSP70 compared with Raf-1 and the inhibitor of the Bag-1 C-terminus primarily influences the interaction of Bag-1 and HSP70 in NP cells. The finding that Bag-1–HSP70 binding was clearly observed in NP cells, but Bag-1–Raf-1 was not under normal conditions suggest the possibility that Bag-1–HSP70 binding enhances HSP70 function to maintain intracellular homeostasis and mitochondrial function in NP cells. These effects are considered to protect NP cells against oxidative stress induced by H_2_O_2_. Moreover, these Bag-1-mediated protective effects suppress p38 phosphorylation in NP cells and downregulate the p38/MAPKs signaling pathway, which enhances NP cell viability. The regulation of p38 activity induces the senescence-associated secretory phenotype [2,37], which reduces NP cell senescence.

With respect to Bag-1 and Raf-1, it has been reported that Bag-1 and Ras each activate Raf-1 independently [15,27]. Subsequently, Bag-1–Raf-1 binding activates signaling molecules including ERK, which attenuates apoptosis [17,25,42]. In contrast, IVD studies indicate that the ERK/MAPK signaling pathway is activated by excessive ROS in IVD cells and promotes IVD degeneration, including extracellular matrix reduction, apoptosis, and cell damage [36,38,39]. Thus, the inhibition of ERK signaling is considered a potential treatment for IVD degeneration as it provides some protection against the adverse effects of TNF-α in the IVD [43]. Meanwhile, hypoxia-induced ERK phosphorylation and ERK activation was shown to be necessary for NP cell survival under hypoxic conditions [44]. Therefore, excessive suppression of ERK is also a disadvantage to NP cell survival. We evaluated NP cells under hypoxic conditions (1% O_2_), which reflects the in vivo environment, although culture conditions may affect the ERK response. When we inhibited the C-terminus of Bag-1 in NP cells, ERK phosphorylation was increased compared with normal NP cells, even under H_2_O_2_-free conditions, whereas Bag-1 overexpression did not increase ERK phosphorylation compared with normal NP cells under H_2_O_2_-free conditions. Although the increase in Raf-1 phosphorylation by H_2_O_2_ treatment showed a similar response in all NP cell groups, treatment with the Bag-1 C-terminal inhibitor did not alter ERK phosphorylation with or without H_2_O_2_ stimulation. There are few studies regarding an independent analysis of Raf-1 in NP cells; however, our findings suggest that the Bag-1–Raf-1 interaction is not so important to ERK activation, and that Raf-1 is regulated by different factors, such as Ras, rather than Bag-1 in NP cells. Based on previous reports of ERK in NP cells, if Bag-1 enhances ERK activation via Raf-1, it is not beneficial for NP cell protection. Therefore, our results, which did not show a Bag-1–Raf-1 interaction, do not contradict the cell protective effects of Bag-1.

The present study suggests the following two possibilities: (1) Bag-1 preferentially interacts with HSP70, rather than Raf-1, to protect NP cells against oxidative stress; (2) the effect of Bag-1 on Raf-1 may be less compared with that of HSP70 in NP cells. These possibilities are supported by the finding that NP cells, which exhibited inhibited Bag-1–HSP70 binding, had decreased cell viability and mitochondrial function under H_2_O_2_ treatment.

We also did not observe significant changes in JNK/MAPKs signaling under the conditions of this study. There are the reports that oxidative stress, such as H_2_O_2_ or tert-butyl hydroperoxide (t-BHP), provoke JNK phosphorylation in disc cells, and HSP70 has been reported to downregulate the JNK/c-Jun pathway [11,12]. These discrepancies presumably result from differences in the types of oxidative stress-inducing agents, administration conditions, and cell culture conditions.

Our study had some limitations. This study focused on the protective ability of Bag-1 and whether Bag-1, HSP70, and Raf-1 interacted in NP cells. Thus, other signaling pathways or factors that have been reported to be involved, such as MAPKs, HSP70, Raf kinases, other Bag-1 interaction partners, such as Bcl-2, and mitochondrial function, were not evaluated in this study. To quantitate and compare the binding of HSP70 or Raf-1 was also difficult. We have concluded that Bag-1-HSP70 contributes to maintaining mitochondrial function; however, there is a possibility that mitochondrial HSP70 (e.g., mtHSP70) interacts with Bag-1, which remains unclear and requires further investigation.

In conclusion, we demonstrated, for the first time, that Bag-1 protects NP cells from oxidative stress by interacting with HSP70. We also showed that Bag-1 primarily interacts with HSP70, rather than Raf-1, in NP cells. Bag-1 suppressed mitochondrial dysfunction and p38/MAPKs activation under oxidative stress conditions, resulting in NP cell survival. However, additional studies are needed to confirm the effects of Bag-1 and its potential interaction with other intracellular signaling factors that may play an important role in understanding the pathogenesis of intervertebral disc degeneration and its treatment.

## Figures and Tables

**Figure 1 biomedicines-11-00863-f001:**
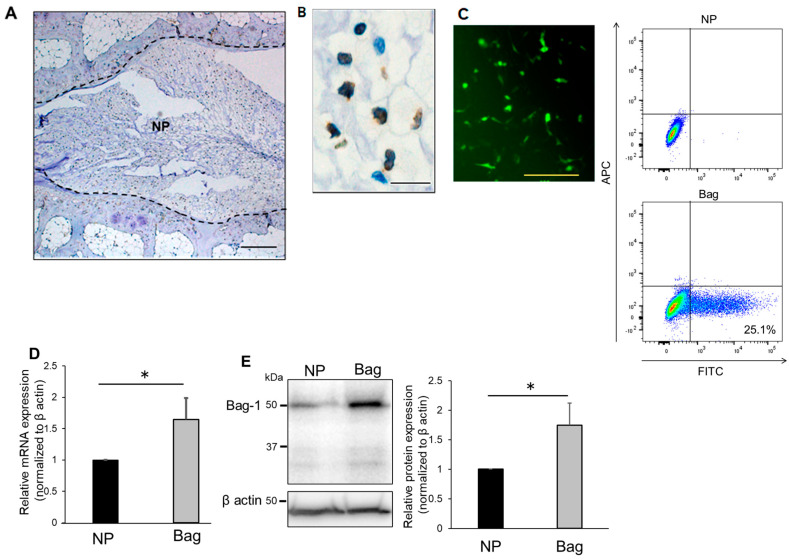
(**A**,**B**) Immunohistochemical staining of Bag-1 in nucleus pulposus (NP) cells of the rat intervertebral disc. Sagittal sections of the mature rat intervertebral disc (**A**) and higher magnification of NP (**B**). Scale bar in (**A**) = 200 μm, scale bar in (**B**) = 20 μm. (**C**) After 48 h of transfection, GFP-positive cells (green) were successfully Bag-1 transfected cells. Representation of flow cytometry analysis showed the average transfection efficiency was 25.1%. Scale bar = 200 μm. NP; NP cells, Bag; Bag-1 plasmid transfected NP cells. (**D**) Measurement of Bag-1 expression by real-time PCR in NP cells and in Bag-1overexpressing NP cells. Data are the means ± SD n = 3. * *p* < 0.05. NP; NP cells, Bag; Bag-1 plasmid transfected NP cells. (**E**) Evaluation of Bag-1 expression by Western blot analysis in NP cells and in Bag-1-overexpressing NP cells. The size of the Bag-1 protein was approximately 50 kDa, and over-expressed Bag-1 protein was a similar size. This indicates that Bag-1L was the main form of the Bag-1 protein expressed in NP cells and the Bag-1 plasmid overexpressed the same protein. Data are the means ± SD n = 3, * *p* < 0.05. NP; NP cells, Bag; Bag-1 plasmid transfected NP cells.

**Figure 2 biomedicines-11-00863-f002:**
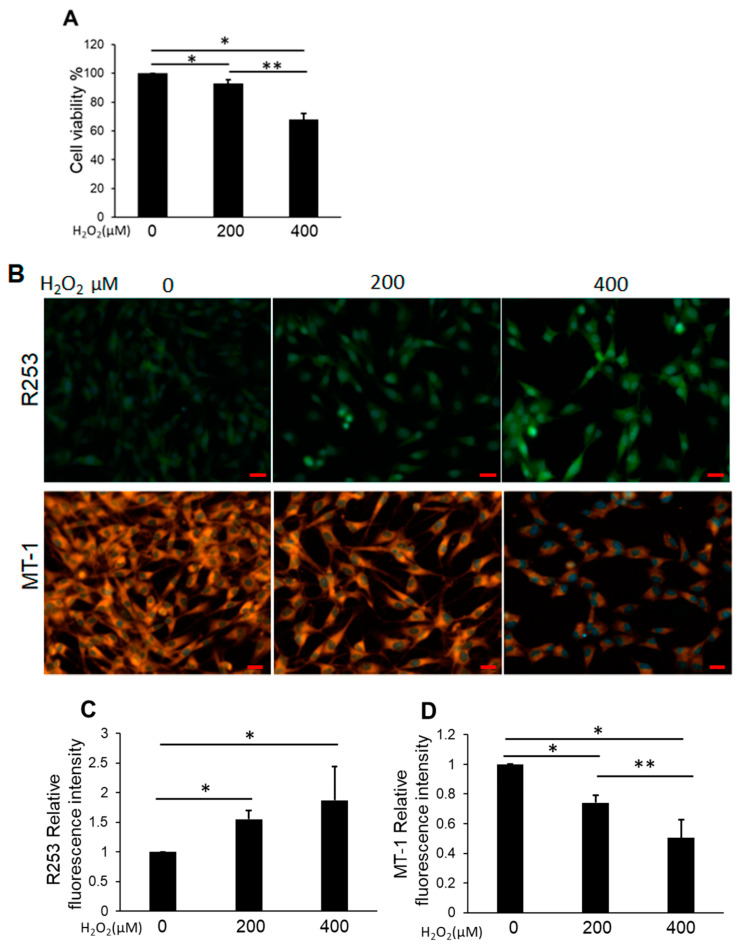
(**A**) Cell viability of H_2_O_2_-treated NP cells. Cells were cultured in DMEM with between 200 and 400 μM H_2_O_2_ for 24 h. Data are the means ± SD n = 4. * *p* < 0.05, ** *p* < 0.05. (**B**) The upper row of the figure represents the ROS production monitored using R253 (green) and the lower row is the mitochondrial membrane potential monitored using MT-1 (orange). The nucleus was detected by Hoechst 33342 reagent (blue). The detection of ROS production increased under H_2_O_2_ treatment, and the detection of MT-1 decreased. Scale bars = 20 μm. (**C**) Quantitative data of ROS production (R253). Data are the means ± SD n = 3, * *p* < 0.05. (**D**) Quantitative data of the mitochondrial membrane potential (MT-1). Data are the means ± SD n = 3, ** p* < 0.05, ** *p* < 0.05.

**Figure 3 biomedicines-11-00863-f003:**
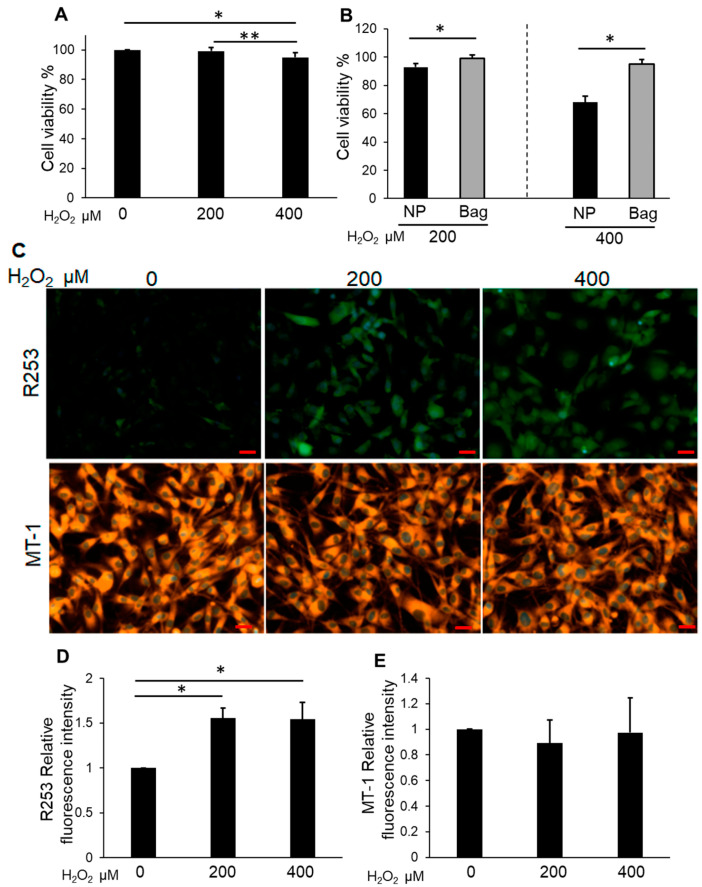
(**A**) Cell viability of H_2_O_2_-treated Bag-1-overexpressing NP cells. Cells were cultured in DMEM with between 200 and 400 μM H_2_O_2_ for 24 h. Data are the means ± SD n = 4, * *p* < 0.05, ** *p* < 0.05. (**B**) Comparison of the cell viability in NP cells (NP) and Bag-1 overexpressing NP cells (Bag) following treatment with H_2_O_2_ for 24 h. Data are the means ± SD n = 4, * *p* < 0.05. (**C**) The upper row of the figure represents ROS production monitored using R253 (green) and the lower row is the mitochondrial membrane potential monitored using MT-1 (orange). The nucleus was detected by Hoechst 33342 reagent (blue). Although the detection of ROS production increased under H_2_O_2_ treatment, the detection of MT-1 was maintained. Scale bars = 20 μm. (**D**) Quantitative data of ROS production (R253) in Bag-1-overexpressing NP cells. Data are the means ± SD n = 3, * *p* < 0.05. (**E**) Quantitative data of the mitochondrial membrane potential (MT-1) in Bag-1- overexpressing NP cells. Data are the means ± SD n = 3.

**Figure 4 biomedicines-11-00863-f004:**
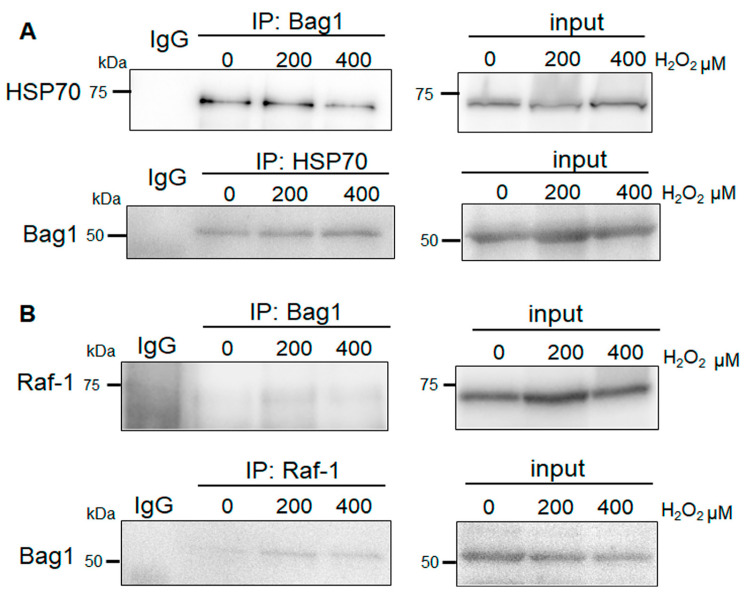
(**A**) Immunoprecipitation (IP) of Bag-1 and HSP70 from NP cells cultured with H_2_O_2_ from 0–400 μM for 24 h followed by Western blot analysis using anti-HSP70 and anti-Bag-1. Bag-1 bound to HSP70 under all conditions. Pre-immune rabbit IgG was used as a negative control for the IP assays. (**B**) Immunoprecipitation (IP) of Bag-1 and Raf-1 from NP cells cultured under H_2_O_2_ from 0–400 μM for 24 h followed by Western blot analysis using anti-Raf-1 and anti-Bag-1 antibodies. Pulldown of Bag-1 did not show obvious co-immunoprecipitation of Raf-1. Pulldown of Raf-1 did not show co-immunoprecipitation of Bag-1 under H_2_O_2_-free conditions and showed an exceedingly faint co-precipitation of Bag-1 under H_2_O_2_ treatment. Pre-immune rabbit IgG was used as a negative control for the IP assays.

**Figure 5 biomedicines-11-00863-f005:**
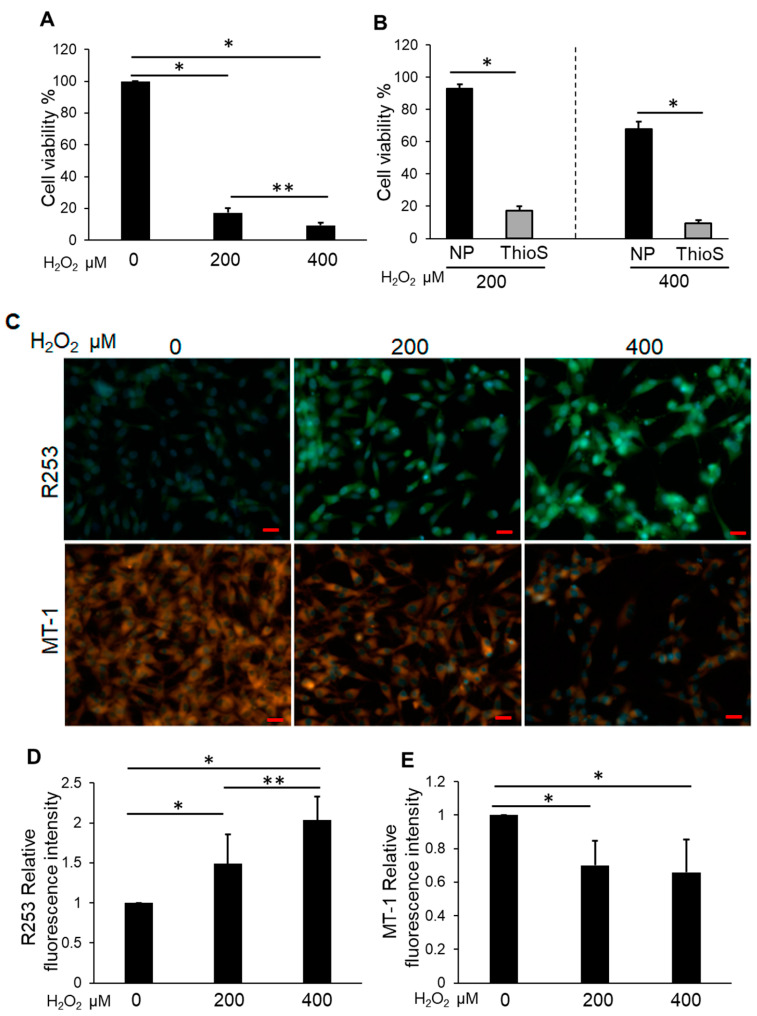
(**A**) Cell viability of thioflavin-S (ThioS)-treated NP cells under H_2_O_2_ treatment. NP cells were treated with 100 μM thioflavin-S for 16 h and then fed fresh DMEM without or with between 200 and 400 μM H_2_O_2_ for 24 h. Data are the means ± SD n = 4, * *p* < 0.05, ***p* < 0.05. (**B**) Comparison of the cell viability of NP cells (NP) and thioflavin-S-treated NP cells (ThioS) treated with H_2_O_2_ for 24 h. Data are the means ± SD n = 4, * *p* < 0.05. (**C**) The upper row of the figure represents ROS production, which was monitored using R253 (green) and the lower row is the mitochondrial membrane potential, which was monitored using MT-1 (orange). The nucleus was detected by Hoechst 33342 reagent (blue). The detection of ROS production increased under H_2_O_2_ treatment, and the detection of MT-1 decreased. Scale bars = 20 μm. (**D**) Quantitative data of ROS production (R253) in thioflavin-S-treated NP cells. Data are the means ± SD n = 3, * *p* < 0.05, ** *p* < 0.05. (**E**) Quantitative data of the mitochondrial membrane potential (MT-1) in thioflavin-S-treated NP cells. Data are the means ± SD n = 3, * *p* < 0.05.

**Figure 6 biomedicines-11-00863-f006:**
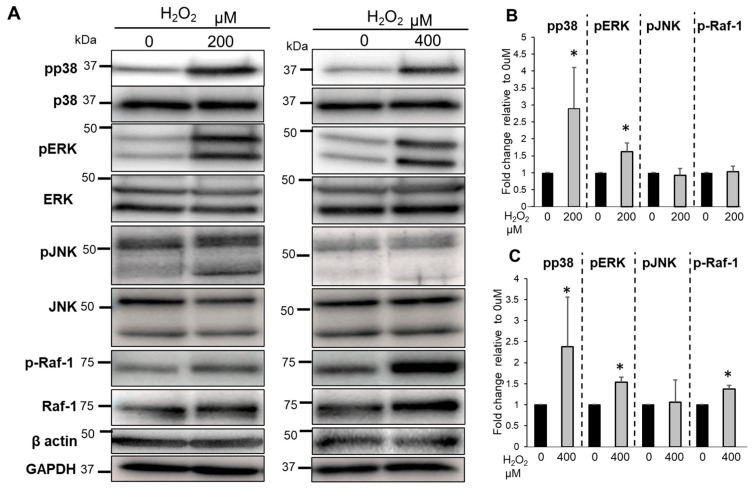
Evaluation of the phosphorylation of MAPK family members and Raf-1 in NP cells. Western blotting (**A**) and quantification (**B**,**C**) of the phosphorylated forms of p38, ERK, JNK, and Raf-1in NP cells treated with or without H_2_O_2_ for 30 min. Data are the means ± SD n = 3, * *p* < 0.05.

**Figure 7 biomedicines-11-00863-f007:**
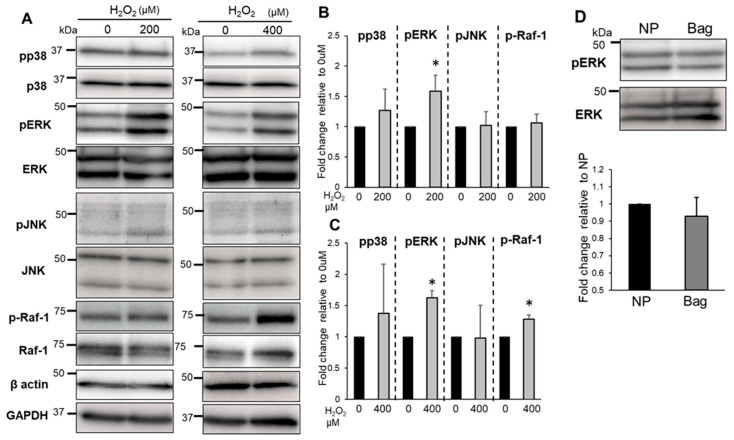
Evaluation of the phosphorylation of MAPK family members and Raf-1 in Bag-1-overexpressing NP cells. Western blot analysis (**A**) and quantification (**B**,**C**) of the phosphorylated forms of p38, ERK, JNK, and Raf-1 in NP cells treated with or without H_2_O_2_ for 30 min. Data are the means ± SD n = 3, * *p* < 0.05. (**D**) Comparison of the expression of pERK in NP cells (NP) and Bag-1-overexpressing NP cells (Bag) without H_2_O_2_ treatment. Data are the means ± SD n = 3, * *p* < 0.05.

**Figure 8 biomedicines-11-00863-f008:**
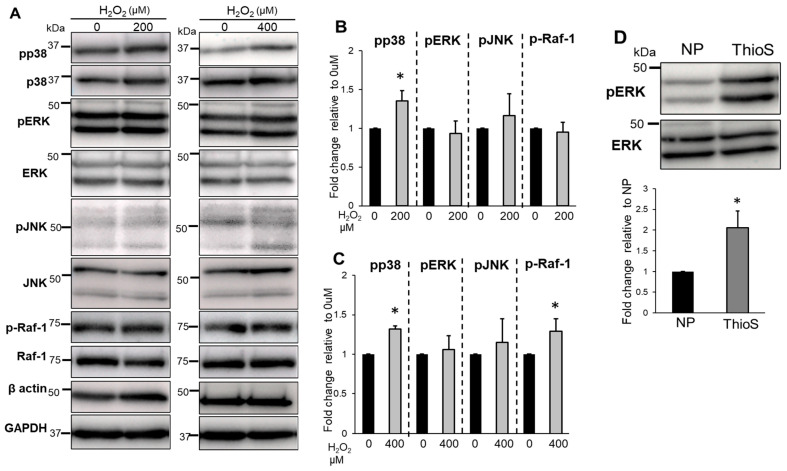
Evaluation of the phosphorylation of MAPK family members and Raf-1 in thioflavin-S-treated NP cells. Western blot analysis (**A**) and quantification (**B**,**C**) of the phosphorylated forms of p38, ERK, JNK, and Raf-1 in thioflavin-S-treated NP cells treated with or without H_2_O_2_ for 30 min. Data are the means ± SD n = 3, * *p* < 0.05. (**D**) Comparison of the expression of pERK in NP cells (NP) and thioflavin-S-treated NP cells (ThioS) without H_2_O_2_ treatment. Data are the means ± S.D. n = 3, * *p* < 0.05.

## Data Availability

Not applicable.

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
