# Peer review of "Bag-1 Protects Nucleus Pulposus Cells from Oxidative Stress by Interacting with HSP70"

_biomedicines, 2023, doi:10.3390/biomedicines11030863_

Round 1

Reviewer 1 Report

In the present study, authors demonstrated the role of Bag-1 in protecting NP cells from H2O2-indiced oxidative stress through its interaction with HSP70. They also showed that Bag-1 is able to suppress mitochondrial dysfunction and to induce p38/MAPKs activation under oxidative stress, thus contributing to maintain NP cell survival. By doing so, authors provide a novel mechanism in understanding the pathogenesis of intervertebral disc degeneration and, potentially, its treatment. 

The study is clearly written, results are well described, methods are appropriate and results support conclusion. Discussion is well organized. 

Author Response

Thank you very much for reviewing our manuscript.

Reviewer 2 Report

              The degeneration of intervertebral disc causes disorders in joints and spine such as pains, disc herniation, spinal canal stenosis and spinal deformalities. In this manuscript, Suyama K et al demonstrated that overexpression of Bag-1 attenuated H2O2-induced oxidative stress in nucleus pulposus cells. This manuscript is well-organized; however, following points should be clarified.

Major points

#1: Please show the transfection efficiency of plasmid (pIRES2-AcGFP1-Bag-1) into NP cells by lipofectamine 3000.

#2: In figure 6-8, the semi-quantifications of signal images were performed. The fold changes are calculated by phosphorylated forms. Did authors use the non-phosphorylated proteins as denominator in each proteins?

Minor points

##1: In figure 1B, the signal of DAB was not clear. Please show more clearly.

##2: In figure 3A, please show the result that describe Bag-1-overexpressing NP cells more clearly.

Author Response

# Comments and Suggestions for Authorsï¼’

The degeneration of intervertebral disc causes disorders in joints and spine such as pains, disc herniation, spinal canal stenosis and spinal deformalities. In this manuscript, Suyama K et al demonstrated that overexpression of Bag-1 attenuated H2O2-induced oxidative stress in nucleus pulposus cells. This manuscript is well-organized; however, following points should be clarified.

Major points

#1: Please show the transfection efficiency of plasmid (pIRES2-AcGFP1-Bag-1) into NP cells by lipofectamine 3000.

Answer: Thank you for kind suggestion. We added the results of pIRES2-AcGFP1-Bag-1 plasmid transfection as the figure of fluorescence microscope dark field and flow-cytometry analysis in Figure1C and Results. The average transfection efficiency was 25.1% at 48h after transfection.

#2: In figure 6-8, the semi-quantifications of signal images were performed. The fold changes are calculated by phosphorylated forms. Did authors use the non-phosphorylated proteins as denominator in each proteins?

Answer: The reviewer’s concern is understandable. As described in Materials and Methods, β-actin or GAPDH were used as loading controls and to normalize protein expression analysis for densitometry quantification. Then, we calculated the fold of phosphorylated forms using the mean value of 0μM of H2O2 in each proteins as denominator. Thus, we did not use the non-phosphorylated proteins as denominator in each proteins.

Minor points

#1: In figure 1B, the signal of DAB was not clear. Please show more clearly.

Answer: We made the signal of DAB more clearly.

#2: In figure 3A, please show the result that describe Bag-1-overexpressing NP cells more clearly.

Answer: We added the significant differences between H2O2 200uM and 400uM in figure 3A and the sentence about that in Result 3.3..
